# Gut Dysbiosis and Diabetic Foot Ulcer: Role of Probiotics

**DOI:** 10.3390/pharmaceutics14112543

**Published:** 2022-11-21

**Authors:** Ankit Awasthi, Leander Corrie, Sukriti Vishwas, Monica Gulati, Bimlesh Kumar, Dinesh Kumar Chellappan, Gaurav Gupta, Rajaraman D. Eri, Kamal Dua, Sachin Kumar Singh

**Affiliations:** 1School of Pharmaceutical Sciences, Lovely Professional University, Phagwara 144411, India; 2Faculty of Health, Australian Research Centre in Complementary and Integrative Medicine, University of Technology Sydney, Ultimo, NSW 2007, Australia; 3Department of Life Sciences, School of Pharmacy, International Medical University, Bukit Jalil, Kuala Lumpur 57000, Malaysia; 4School of Pharmacy, Suresh Gyan Vihar University, Mahal Road, Jaipur 302017, India; 5Department of Pharmacology, Saveetha Dental College, Saveetha Institute of Medical and Technical Sciences, Saveetha University, Chennai 602105, India; 6Uttaranchal Institute of Pharmaceutical Sciences, Uttaranchal University, Dehradun 248007, India; 7School of Health Sciences, The University of Tasmania, Launceston, TAS 7248, Australia; 8Discipline of Pharmacy, Graduate School of Health, University of Technology Sydney, Ultimo, NSW 2007, Australia

**Keywords:** diabetic foot ulcer, pathogenesis, sources of probiotics, therapeutic potential of probiotics on DFU, market status of probiotics, patents on probiotics

## Abstract

Diabetic foot ulcer (DFU) is a multifactorial disease and one of the complications of diabetes. The global burden of DFU in the health sector is increasing at a tremendous rate due to its cost management related to hospitalization, medical costs and foot amputation. Hence, to manage DFU/DWs, various attempts have been made, including treating wounds systematically/topically using synthetic drugs, herbal drugs, or tissue engineering based surgical dressings. However, less attention has been paid to the intrinsic factors that are also the leading cause of diabetes mellitus (DM) and its complications. One such factor is gut dysbiosis, which is one of the major causes of enhancing the counts of Gram-negative bacteria. These bacteria produce lipopolysaccharides, which are a major contributing factor toward insulin resistance and inflammation due to the generation of oxidative stress and immunopathy. These all lead to DM and DFU. Probiotics are the commercial form of beneficial gut microbes that are taken as nutraceuticals by people of all ages to improve gut immunity and prevent gut dysbiosis. However, the role of probiotics has been less explored in the management of DFU. Hence, the therapeutic potential of probiotics in managing DFU is fully described in the current review. This report covers the linkage between gut dysbiosis and DFU, sources of probiotics, the mechanisms of probiotics in DW healing, and the impact of probiotic supplementation in treating DFU. In addition, techniques for the stabilization of probiotics, market status, and patents related to probiotics have been also covered. The relevant data were gathered from PubMed, Scopus, Taylor and Francis, Science Direct, and Google Scholar. Our systematic review discusses the utilization of probiotic supplementation as a nutraceutical for the management of DFU.

## 1. Introduction

Diabetic foot ulcer (DFU) is the one of the most common complications of diabetes. The global prevalence of DFU due to diabetes is 25%. It is an open sore wound that occurs in the foot. It generally occurs due to the hypoxia and oxidative stress caused by reactive oxygen species, a decrease in the level of growth factors (GFs), nucleic acids and the lack of glycemic control. DFU has reached the 10th position in terms of the annual economic burden of diabetics [1]. this situation has arisen because of a lack of existing treatment strategies to promote wound healing. In DFU, delayed wound healing occurs [2]. The common reason for this is the extended inflammatory response that leads to impairment in keratinocyte migration, collagen synthesis, vascularization, fibroblast migration, epithelialization, collagen proliferation, differentiation and migration. Overall, these contributing factors often result in amputation and even the death of the DFU patient. The global prevalence of amputation due to DFU in 2022 is reported to be 10–15% [3].

The treatment of DFU is challenging, as it involves multiple stages, etiologies and degrees of severity that vary among the diabetic mellitus (DM) patients. The existing formulations on the market provide adequate glycemic control. However, these are unable to treat the various stages of DFU in DM patients. Therefore, this increases the burden of medications on patients suffering to DFU, because the delay in wound healing may also be dependent on the severity of the wound, rather than only glycemic control. Hence, for wound healing, the administration of antibiotics or anti-inflammatory agents is also required. Other approaches that are used to manage DFU include plastic surgery, orthopedics, vascular surgery, offloading, antibiotics (ciprofloxacin, vancomycin, clindamycin and piperacillin/tazobactam), herbal drugs (curcumin, quercetin, aloe vera, achlefan and panchavalkla), synthetic drugs (mevastatin, simvastatin, naltrexone and azelnidipine), growth factors (GFs), nucleic acids gene based delivery, novel drug delivery systems (NDDSs) such as nanostructured lipid carriers, nanoemulsion, nanoparticles and dressings such as gauze, films, foams or, hydrocolloid-based dressings as well as polysaccharide- and polymer-based dressings etc. The limitation of surgery is that in DM patients, there is a slow progression of wound healing. Once the patient has undergone surgery, the wounds take a long time to heal, leaving the patient susceptible to infections. The limitation of synthetic and herbal drugs is their poor solubility and permeability, while the limitations of GFs and nucleic acid are their high cost and low stability. The limitation associated with the NDDS is their low retainability at the injured site, if used topically; additionally, to enhance their retention, they have to be further incorporated into nanomaterials, which increases the cost of therapy. Dressings which are currently available to manage DFU have some limitations, such as the inability to absorb the exudate and high cost. Antibiotics can decrease microbial load but not heal the wound [1,2,3]. These treatment strategies are expensive and underline the need for a multi-disciplinary, cost-effective approach to control hyperglycemia with the potential to target different stages of DFU. In recent years, probiotics have gained tremendous attention for the management of various metabolic diseases due to their anti-infective, antioxidant, anti-inflammatory, anti-diabetic and immunomodulatory activities. In the case of DFU, probiotics help to maintain the levels of short chain fatty acids, gut hormones and the endocannabinoid system that helps in maintaining glucose homeostasis, decreasing inflammation and providing immunity to the DFU patients. Probiotics are part of various food products that are consumed on a daily basis. They help to manage gut microbiota function and impart immunomodulation. They also have a commercial status in the form of probiotic drinks and foods [4]. Despite having such potential, they have been clinically less explored for their potential in the management of DFU.

This review comprehensively describes the role of probiotics as multi-disciplinary agents in overcoming the clinical challenges of existing treatment strategies for DFU. Further, this review expounds on the various sources of probiotics, their mechanistic effects on DFU, stabilization techniques and relevant clinical studies, along with filed/granted patents.

## 2. Pathogenesis of Diabetic Wounds

During hyperglycemia, the levels of micro-ribulose nucleic acid (miR)-155, miR-191, miR-200b, miR-15b, miR-200, and miR-205–5p are increased while those of miRNA-146a and miR-132 are decreased. The overactivation of miR-155, miR-191 and miR-200b results an increase in the level of myeloperoxidase (MPO)-positive cells and C-reactive protein levels, which, in turn, leads to impairment in angiogenic markers such as collagen 1, transforming growth factor (GF) beta-1 and alpha-smooth muscle actin. In addition, they prolong the inflammatory phase of wound healing and impede the wound healing process. Besides these factors, the overactivation of miR-15b, miR-200 and miR-205–5p results in the impairment of the vasoendothelial GF pathways and impedes the wound healing process. The decrease in the levels of miRNA-146a and miR-132 activates the tumor necrosis factor receptor-associated factor 6 (TRAF6), interleukin-1 receptor associated kinase 1 (IRAK1) and toll-like receptors. The overactivation of these pathways results in an increase in the level of inflammatory markers that prolongs the inflammatory phase and delays the wound healing process [3]. In addition to this, in DFU, the level of matrix mettalo proteinase (MMP) also gets increased, which inhibits the migration of keratinocytes toward the wound site and impairs collagen synthesis. This delays the wound healing process [1].

High blood glucose levels also result in idiopathic complications, viz. neuropathy, immunopathy and vasculopathy. Neuropathy affects sensory, motor and autonomic nerves. In sensory neuropathy, there is a loss of pain leading to unnoticed trauma, which, in turn, may lead to ulcer formation. In motor neuropathy, weakness and wasting of intrinsic foot muscles occur, which results in abnormal gait and foot deformities that can lead to ulceration. In autonomic neuropathy, sweat glands get suppressed, which results in a decrease in the sweating rate at the foot site. This makes the skin dry and brittle and leads to secondary infections and, finally, ulceration. Vasculopathy is a general term used to describe any disease affecting blood vessels. It is generally of two types: microanginopathy and macroanginopathy. Microanginopathy occurs when there is deposition of glycoproteins and blood clots on the surface of the basement of the vessels. This deposition makes the walls of the vessels thicker and causes leakage from them, leading to ulceration. Macroanginopathy includes the deposition of fats and blood clots in the blood vessels. This decreases the blood flow in the vessels, which leads to necrosis and, finally, ulceration. In the case of immunopathy, there is a decrease in immunity due to the decrease in the level of polymorpholeukocytes, intracellular killing rate and GFs, coupled with an excess of metalloproteinases. This prolongs the inflammatory phase and delays the wound healing process (Figure 1A) [2].

## 3. Gut Dysbiosis and DW

During hyperglycemia, there is an imbalance between Gram-positive and Gram-negative bacteria, which leads to gut dysbiosis. Imbalance in the gut microbiome ultimately results in alterations in the synthesis of short chain fatty acids (SCFA) and the secretion of gut hormones (GLP-1 and PYY). This imbalance increases the level of lipopolysaccharides (LPS) in the systemic circulation, impairs bile acid metabolism and alters circulatory branched-chain amino acids. Alterations in the SCFAs levels and gut hormones result in impairment in glucose homeostasis and lipids. Increase in the level of LPS results in metabolic endotoxemia, activates toll like receptors and causes inflammation by promoting the secretion of pro-inflammatory cytokines. Moreover, impairment in bile acid metabolism inhibits the conversion of primary bile acids such as cholic and chenodeoxycholic acids into secondary bile acid species. i.e., deoxycholic and lithocholic acids. This results in the dysregulation of glucose homeostasis. Alterations in circulating branched-chain amino acids lead to a decrease in the level of GLP-1 and impair glucose homeostasis. In addition, gut dysbiosis also diminishes the endocannabinoid system and impairs the inflammatory and immunomodulatory responses of the body. Overall, these factors result in impaired glucose homeostasis and immunity and an increase in inflammation, all of which are key contributors to DFU. To address gut dysbiosis, probiotics are suitable candidates due to their numerous health benefits (Figure 1B) [2,3,4].

## 4. Sources of Probiotics

Rich sources of probiotics are dairy and dairy-related products [5]. Micro-organisms, such as bifidobacteria and lactic acid bacteria (LAB), are extracted from fermented milk and have been used for centuries. It has been found that the fermented milk from Chinese yak, known as kurut, consists of 148 strains of LAB. Among these strains, Streptococcus thermophilus and Lactobacillus delbrueckii subsp bulgaricus are the most prevalent. In addition, Koumiss, Kefir grains and Masai milk are fermented milk items from which lactobacillus strains and yeast with probiotic properties may be obtained [5]. Other sources of probiotics are given in Table 1.

## 5. Therapeutic Potential of Probiotics in Treating DW

DW is associated with oxidative stress, inflammation and immunopathy. Hence, probiotics can play a major role in the therapy of DW. Probiotics have multiple therapeutic actions, such as antioxidant, anti-inflammatory, immunomodulatory and antidiabetic (Figure 1C) [8]. Probiotics exert antioxidant effects by decreasing the oxidative stress generated by mitochondrial dysfunction and reactive oxygen species. It is known that SOD has a short half-life and low bioavailability. They enhance the antioxidant effect by releasing antioxidant enzymes such as SOD and catalase. In mitochondrial dysfunction, oxidative stress is produced by the generation of superoxide reactive oxygen species. When probiotics are consumed, SOD enzymes are produced that help in the breakdown of superoxide ions into hydrogen peroxide and water, thereby decreasing oxidative stress. Therefore, probiotics are suitable for the local delivery of SOD in bowel-related disease. In addition, probiotics also produce catalase enzymes that help in cellular antioxidant defense and promote the decomposition of hydrogen peroxide, which, in turn, inhibits the production of hydroxyl radicals by Fenton reaction. Probiotics also produce antioxidant metabolites such as glutathione butyrate and folate. These metabolites eliminate hydrogen peroxide, peroxynitrite and hydroxyl radicals with the help of selenium-dependent glutathione peroxidase enzyme and reduce oxidative stress [9].

Nuclear factor-kappa B (NF-ĸB) is a key signaling channel which is responsible for inflammation. It is present in the cytoplasm in an inactive form, bound to an inhibitory molecule, i.e., IĸB. During inflammation, IĸB molecule breaks down, which results in the release of NF-ĸB to activate the inflammatory cascades. A probiotics strain such as *Lactobacillus rhamnosus* GG or *Lactobacillus casei* DN-114 001 inhibits the breakdown of the inhibitory molecule- IĸB and reduces the expression of proinflammatory cytokines such as IL-8. In addition, probiotics trigger toll-like receptors, which initiate beta-defensins and exert anti-inflammatory actions [10].

Probiotics exert immunomodulatory actions by interacting with antigen presenting and release chemical mediator cytokines such as interleukins (ILs), tumor necrosis factor, interferons, transforming GF and chemokines from immune cells (lymphocytes, granulocytes, macrophages, mast cells, epithelial cells, and dendritic cells (DCs)), which further regulate the innate and adaptive immune system. In addition, probiotics help in enhancing the production of cytokines, activate the tight junctions of the intestinal barrier against intercellular bacterial invasion, encourage the secretion of immunoglobulin A and production of antibacterial substances and compete with new pathogenic microorganisms for enterocyte adherence. Through these processes, probiotics regulate intestinal epithelial health. An early, innate immune response is also induced by probiotics through phagocytosis, polymorphonuclear (PMN) cell recruitment and tumor necrotic factor-alpha production [11].

Probiotics have an anti-diabetic effect because they help in the production of SCFA, which enhances the release of incretin hormones that influence glucose levels. In addition, probiotics reduce the level of LPS, making them useful for the treatment of gut dysbiosis and type 2 diabetes mellitus. Probiotics also help to increase the levels of GLP-1 and insulinotropic hormones in enteroendocrine L-cells [12]. This optimizes glucose metabolism, reduces cell damage and improves insulin sensitivity. Among several animal models used for DM, it has been reported in 91 research papers that probiotics prevent DM onset by down-regulating certain inflammatory cytokines, such as interferons (IFN) and IL-2 or IL-1, or by increasing anti-inflammatory IL-10 production. It is also claimed that probiotics produce a defensive wall that prevents pathogenic bacterial species from colonizing the epithelium [13].

Studies related to the antioxidant, anti-inflammatory, immunomodulation and anti-diabetic property of probiotics are depicted in the Table 2.

With regard to the therapeutic potential of probiotics, various studies have been carried out in the field of DW healing, which are discussed below.

In one of these studies, Peral et al. (2010) investigated the effect of *Lactobacillus plantarum* against chronic infected leg ulcers in diabetic patients. In their trial, 14 diabetic and 20 non-diabetic patients having venous leg ulcers were considered. For the treatment, topically Lactobacillus plantarum was applied to both diabetic and non-diabetic patients with venous leg ulcers. After 30 days of topical treatment with *Lactobacillus plantarum*, it was observed that 43% of diabetics and 50% of non-diabetic patients showed complete wound healing. Therefore, it was concluded that Lactobacillus plantarum accelerated wound healing in diabetic and non-diabetic patients by exerting antibacterial and anti-inflammatory actions, reducing apoptotic, neutrophils, and necrotic cells and modifying IL-8 production [40].

In another study, Majid et al. (2016) examined the effect of *Lactobacillus casei* and its exopolysaccharide against DW in induced male Wistar diabetic rats. The results revealed that the topical application of *Lactobacillus casei* and its exopolysaccharide showed 1.4-fold and 1.1-fold increase in wound contraction within 14 days as compared to negative and control groups [41].

Similarly, Mohseni et al. (2018) investigated the effect of probiotic supplementation on metabolic status and wound healing in patients with DFU. They performed a double-blind, randomized and placebo-controlled trial. In their trial, 60 patients aged 40–85 years old and having grade 3 (deep ulcer with cellulitis) DFU were considered. These 60 patients were casually distributed into two groups (30 patients on each side) to receive either placebo or oral probiotic capsule (*Lactobacillus fermentum*, *Lactobacillus casei*, *Lactobacillus acidophilus*, and *Bifidobacterium bifidum*) every day for 12 weeks. The dose of the probiotic capsule was 2 × 10^9^ CFU/g each. After 12 weeks, it was observed that compared to the placebo group, the probiotics-treated groups showed a significant reduction in ulcer length (−1.3 ± 0.9 cm for probiotic vs. −0.8 ± 0.7 cm for placebo, *p* = 0.01), ulcer width (−1.1 ± 0.7 cm for probiotic vs. −0.7 ± 0.7 cm for placebo, *p* = 0.02) and ulcer depth (−0.5 ± 0.3 cm for probiotic vs. −0.3 ± 0.3 cm for placebo, *p* = 0.02). Moreover, it was also observed that probiotics not only reduced the ulcer length, size and depth, but also helped in the downregulation of blood glucose level, total serum cholesterol, high sensitivity C-reactive protein (hs-CRP), malondialdehyde (MDA) levels, augmented plasma nitric oxide (NO) and total antioxidant capacity (TAC), indicating the potential of probiotics in treating DFU [42].

In another study, Gonzalez et al. (2018) explored the effect of clindamycin/cefotaxime and *Lactobacillus acidophilus* against micro-organisms isolated from the foot of DFU patients. The turbidimetric method was used for the bioassay. Three types of bacteria were isolated from DFUs strain, i.e., strain 1 (*Pseudomonas* sp.), strain 2 (yeast-like cell) and strain 3 (*Enterobacter* sp.). Then, clindamycin/cefotaxime and *Lactobacillus acidophilus* were tested against micro-organisms isolated from the foot of DFU patients. Clindamycin was used against all the strains isolated from DFU patients at concentrations of 0.15 μg/mL, 0.25 μg/mL, and 50 μg/mL. It was observed that clindamycin was only effective against strain three; the percentages of inhibition were 18, 88, and 89, respectively. Meanwhile, cefotaxime at concentrations of 0.15 μg/mL, 0.25 μg/mL, and 50 μg/mL showed an effect against all the three strains. The percentages of inhibition of cefotaxime at a dose of 0.15 μg/mL against strains 1, 2 and 3 were 85, 70 and 55, respectively. At a dose of 0.25 μg/mL cefotaxime showed a good percentage of inhibition against strains 1, 2 and 3, i.e., 87, 68, and 60, respectively. At a dose, 50 μg/mL cefotaxime showed percentages of inhibition for strains 1, 2 and 3 of 88, 65 and 76, respectively. When *Lactobacillus acidophilus* was tested against all these at concentrations of 40 mg/mL, 400 mg/mL, and 800 mg/mL, it was observed that it was only effective against strains 1 and 3. For strains 1 and 3, *Lactobacillus acidophilus* showed percentages inhibition of 3% and 9%, respectively, at a dose of 40 mg/mL. At dose of 400 mg/mL, *Lactobacillus acidophilus* showed percentages of inhibition against strains 1 and 3 which of 34 and 18, respectively. Similarly, at a dose of 800 mg/mL, *Lactobacillus acidophilus* showed 40% inhibition for strain 1 and 26% inhibition for strain 3, indicating the antibacterial potential of probiotics against the micro-organisms that are responsible for DFU [43].

Similarly, the effect of *Lactobacillus plantarum* gel was evaluated against burns associated DW healing in mature male Sprague-Dawley rats. The results revealed that the topical application of *Lactobacillus plantarum* accelerated DW healing as compared to other treated groups due to its anti-inflammatory action, increased hydroxyproline content, epithelization and angiogenesis at the site of injury [44].

In a related, Venosi et al. (2019) studied the effect of a multi-strain probiotic formulation on infected chronic ischemic wounds. This study was conducted on an 83-year old woman with a history of DM, hypertension and ischemic heart disease. The patient had critical limb ischemia and a cutaneous ulcer on the right leg. In addition, this patient was also subjected to percutaneous transluminal angioplasty (PTA) with a drug eluting balloon (DEB) ranger 5 × 100 mm in the superficial femoral artery (SFA) and right popliteal artery, followed by surgical curettage of necrotic forefoot injuries and amputation of the second toe of the right foot. To manage this, in the initial stage of treatment, the patient was given piperacillin/tazobactam 4.5 g intravenously (I.V) every eight hours. This treatment was given to the patients for 8 days. After that time, a reduction in inflammatory markers was observed, and piperacillin/tazobactam was switched to oral minocycline tablet (100 mg) every 12 h for 15 days. The patient was discharged after 21 days of hospitalization. Then local dressings and polymeric membrane (PolyMem^®^-Ferries Mfg) were applied at the site of injury. In spite of these treatments, the condition of the injury worsened and the patient was referred to the Department of Public Health and Infectious Diseases, University of Rome. His injury was properly examined, and multiple micro-organisms such as *Proteus mirabilis*, *Entero faecalis* and *Klebsiella pneumonia* were isolated. After the identification of these microorganisms, topical 10% cutaneous-iodopovidone solution (Poviderm^®^ 10% Skin Solution) was applied. This treatment led to an improvement in wound healing. Then systemic and topical antibiotics treatment was stopped. Afterwards, it was decided to start treatment with a multi-strain probiotic formulation. The multi-strain probiotic formulation was comprised of lyophilized powder sachets, each containing 100 billion colony forming units (CFU) of *Lactobacillus acidophilus* NCIBMB 43030 20% in weight, *Lactobacillus plantarum* NCIBMB 43029 20% in weight, and *Streptococcus thermophilus* NCIMB 30438 40% in weight. The probiotic treatment was continued for 24 days. The results revealed that the topical application of probiotics at the site of injury led to the inhibition of multiple micro-organisms (*Proteus mirabilis*, *Entero faecalis* and *Klebsiella pneumonia*) and completely healed the wound [45].

Similarly, Chuang et al. (2019) studied the effect of *Lactobacillus plantarum* TWK10-fermented soymilk against DW in male Wistar diabetic rats. The results revealed that the topical application of *Lactobacillus plantarum* TWK10-fermented soymilk accelerated DW healing within 14 days by promoting collagen deposition and angiogenesis, increasing hydroxyproline content and decreasing oxidative stress, as well as by its antimicrobial action at the site of injury [46].

In another study, Kumari et al. (2019) examined the effect of *Streptococcus thermophilus* and low-level laser therapy on DW healing in male Albino diabetic rats. The results revealed that the topical application of saline did not lead to effective wound contraction while *Streptococcus thermophilus* showed a reduction in oxidative stress and promoted DW healing. However, it was observed that when *Streptococcus thermophilus* treatment and low-level laser therapy were used in combination, accelerated DW healing occurred. In addition, the combination promoted angiogenesis and collagen deposition at the site of injury [47].

Similarly, the effect of probiotics supplementation on DW healing was tested in male adult Wistar rats. In this study, 46 rats were used, divided into two groups, i.e., control and probiotic-treated groups. The latter received Probiatop^®^, while the control group received maltodextrin. The oral daily dose of both supplements was 250 mg once a day. Then, each group was further subdivided into two subgroups on the basis of euthanasia: 3rd or 10th postoperative (PO, subgroups C3 = 12 rats, P3 = 12 rats, C10 = 11 rats, P10 = 11 rats). Diabetes was induced to all rats by inducing alloxan. Supplementation was started five days before surgery and continued until euthanasia. The results revealed that the P10 group showed maximal wound contraction as compared to the C10 group. It was also observed that from the 3rd to 10th post-operative day, the probiotic treated group showed an increment in type 1 collagen deposition at the site of injury as compared to the control group. Hence, it was concluded that probiotic supplementation accelerated DW healing in rats by enhancing neovascularization and collagen deposition at the site of injury [48].

Similarly, Layus et al. (2020) studied the antibacterial activity of a probiotic containing *Lactobacillus plantarum* CRL 759 against microorganisms Pseudomonas aeruginosa and methicillin-resistant *Staphylococcus aureus* (MRSA), isolated from the foot of a DFU patient. The antimicrobial activity of the probiotic was determined by different methods, such as the modified agar slab method and the agar well diffusion method. The outcomes showed that *Lactobacillus plantarum* CRL 759 sans cell supernatant (SLp759) restrained both MRSA and *Pseudomonas aeruginosa* development. Likewise, SLp759 repressed the grip of pathogenic organisms. Furthermore, after the balance of acidic SLp759, no action against micro-organism strains was observed. In addition, treatment with proteolytic chemicals did not adjust antibacterial movement, demonstrating that no bacteriocin was available in the supernatant. Additionally, the results obtained by HPLC examination demonstrated that the inhibitory impact was the aftereffect of the creation of two natural acids, i.e., lactic and acetic [49].

In another study, Mohtashami et al. investigated the effect of Lactobacillus *Plantarum* against DW in alloxan-induced male Wistar diabetic rats. The results revealed that the *Lactobacillus plantarum* treated groups exhibited 1.14- and 1.35-fold increases in wound closure within 14 days in comparison to *Lactobacillus bulgaricus* and diabetic control-treated groups. In addition, the *Lactobacillus plantarum* treated groups showed accelerated DW healing due to the anti-inflammatory action, cell migration and proliferation at the site of injury [50].

## 6. Techniques Used for the Stabilization of Probiotics

Despite having various pharmacological as well as health benefits, probiotics are less commercialized due to their degradation upon exposure to sunlight, low pH, high temperatures and oxygen. It has been found that bacteria such as LAB excrete polysaccharides (EPS) that provide protection against harsh conditions. However, this protection is not sufficient. The different approaches used by the researchers to improve the stability and survival of probiotics include culture pre-exposure to the sub-lethal stresses [51] and the incorporation of micro-nutrients such as two-step fermentation [52], microencapsulation [53], the use of oxygen-impermeable containers [54] and immobilization [55]. Among these techniques, microencapsulation is the most widely used by researchers.

Microencapsulation is the process of packaging solids, liquids or gases into miniature containers. It increases stabilization and the survival rate of the probiotics at the time of processing, prevents oxidative reactions, provides sustained release at a target site and enhances shelf life [53]. Microencapsulation may be categorized into chemical and physical techniques. Both play a key role in the pharma and food sectors. Forms of physical encapsulation include spray chilling [56], suspension coating [57], fluidized bed coating [58], liposome entrapment [59], centrifugal extrusion [60], spray cooling [61], rotational suspension separation [62], annular jet, spray coating [60], spinning disk [63], air spray drying extrusion coating [60] and pan coating [64]. Chemical methods include in situ polymerization [57], interfacial polymerization [65], matrix polymerization [57] and extrusion [57]. Numerous studies on the microencapsulation technique have shown that emulsions are commonly used to enclose probiotic cultures within solid fat microcapsules, helping them to retain their vitality and activity. It is well-known that powdered foods have longer shelf-lives at normal room temperatures. Techniques that are used to dry probiotics to enhance their stability include microwave drying, spray drying, vacuum drying and lyophilization [60]. Among these, lyophilization is the best technique to maintain the viability of bacterial cells in order to use them in the preparation of starter culture cells. In addition to this, materials used for encapsulating probiotic strains include pectin [66], locust bean gum [67], rennet [68], whey protein [66], cellulose [69], к-carrageenan [70], chitosan [71] and alginate [57]. These materials act as gelling agents or support materials in the probiotic strain encapsulation. Various efforts made by the researchers to improve the stability of probiotics are listed in Table 3.

The advantages and disadvantages of commonly used techniques for the stabilization of probiotics [66,70] are discussed below.
Freeze drying—Advantages: (i) Easy and convenient; (ii) Does not require freezing conditions. Disadvantages: Lengthy and expensive.Spray drying—Advantages: (i) Fast drying process; (ii) Powdered material obtained directly; (iii) Simple and easy to alter drying conditions; (iv) High production efficiency. Disadvantages: (i) Costly; (ii) An excessive amount of air is needed to increase the power consumption; (i) Equipment is complex; (ii) C overs large area.Fluidized bed dryer—Advantages: (i) High thermal efficiency; (ii) Handling time is short; (iii) It is possible to the materials in a shorter time. Disadvantages: (i) Chance of attrition of materials; (ii) Many organic powders develop electrostatic charge during drying.Extrusion—Advantages: (i) Low cost; (ii) Flexible. Disadvantages: (i) Size variances; (ii) Product limitationMicroencapsulation—Advantages: (i) Protects materials from external stress; (ii) It is possible to prepare sustained and controlled release formulations. Disadvantages: (i) High cost; (ii) Non uniform coating effect the release profile of the active moiety in the body.


## 7. Market Status of Probiotics

The health benefits and pharmacological actions of probiotics have been gaining the attention of consumers. The global market for probiotics is divided into different categories, i.e., dietary supplements, drinks, foods and animal feeds. Probiotic food may be further subdivided into baby food, yogurt, infant formula, breakfast cereals/baked goods and other probiotic foods. Additionally, probiotic drinks may be further classified into fruit-based and dairy-based drinks. Regarding distribution channels, the market for probiotics may be segmented into convenience stores, hypermarkets/supermarkets, pharmacies and drug stores, online channels and other distribution channels. In addition, the probiotics market is projected to register a CAGR rate of 7.2% during the forecast period of 2020–2030 [86]. Countries and regions which have become hubs of the probiotics market include North America (USA, Mexico, and Canada), Europe (Russia, Spain, UK, France, and Italy), Asia-Pacific (China, India, Japan, and Australia), South America (Argentina, Brazil) and the Middle East and Africa (Saudi Arabia, South Africa) [87]. Lists of probiotics that are available on the global market and patents on probiotics are depicted in Table 4, Table 5 and Table 6.

## 8. Conclusions

The data gathered in this review suggest that the oral consumption and topical application of probiotics bring about remarkable improvements in DFU. Moreover, the oral consumption of probiotics is much better than topical application. This is because oral probiotics have the ability to colonize the gut microbiota and improve gut dysbiosis by exerting anti-inflammatory, immunomodulatory, antioxidant and antidiabetic effects, which is restricted in topical application. The topical route will only provide a local effect decreasing the microbial load at the site of injury. Numerous preclinical as well as in vitro studies have shown the therapeutic potential of probiotics against DFU. Despite these enormous potentials, these studies are confined to academic laboratories. There are limited clinical studies on the use of probiotics against DFU. One of the leading reasons for this is the complexity in the identification and isolation of the probiotics, as well as their poor stability and high cost. Therefore, more clinical-based research is required to augment the pharmacotherapeutic potential of probiotic supplementation. Further, from a commercial perspective, it is important to seek novel techniques to enhance the stability of probiotics. Understanding the aforementioned bottlenecks and finding novel strategies to overcome them may bring about novel, effective treatments for DW.

## Figures and Tables

**Figure 1 pharmaceutics-14-02543-f001:**
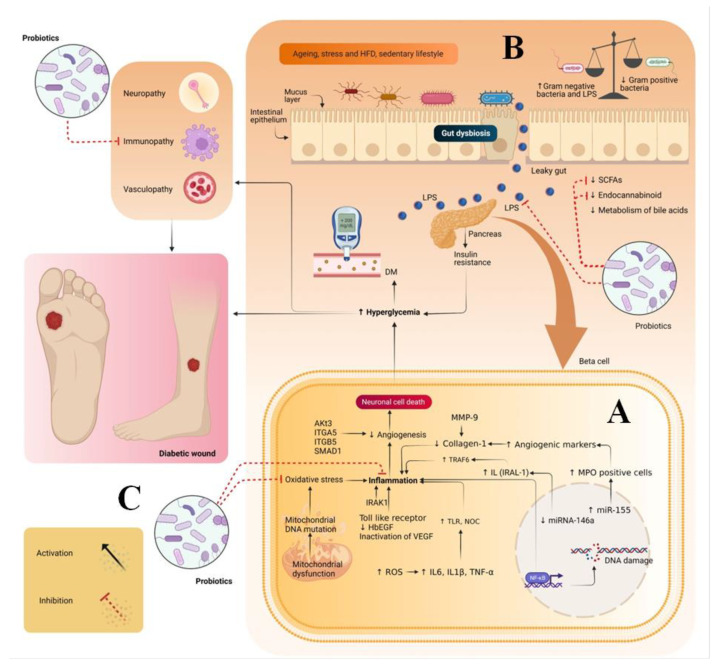
(**A**) Pathogenesis of DFU (**B**) Gut dysbiosis and its relation with pathogenesis of DFU and (**C**) the role of probiotics in the treatment of DFU. ↑ indicates upregulation and symbol ↓ indicates downregulation.

**Table 1 pharmaceutics-14-02543-t001:** Fruit and vegetable-based source of probiotics [5,6,7].

Source	Fermented Product	Micro-Organism Isolated
Bamboo shoots	Soidon	*Lactococcus lactis*, *Lactobacillus brevis* and *Leuconostoc fallax*
Black mustard seeds	Hardline	*Lactobacillus sanfranciscensis*, *Lactobacillus casei*, *Lactobacillus brevis*, *Lactobacillus acetotolerans*, *Lactobacillus paracasei* and *Lactobacillus pontis*
Broccoli	Yan-tsai-shin	*Leuconostoc Mesenteroides*, *Weissella cibaria*, *Lactobacillus plantarum*, *Enterococcus sulfurous* and *Weissella paramesenteroides*,
Cabbage	Dhamuoi	*Leuconostoc mesenteroides* and *Lactobacillus plantarum*
Celery, cabbage, radish, and cucumber	Pascal	*Lactobacillus brevis*, *Lactobacillus plantarum*, *Lactobacillus lactis*, *Leuconostoc mesenteroides*, *Lactobacillus fermentum*, and *Lactobacillus pentosus*
Cherries	Cherries juice	*Enterococcus gallinarum* and *Pediococcus pentosaceus*
Chinese cabbage	Kimchi	*Weissella koreensis*, *Lactobacillus lactis*, *Lactobacillus plantarum*, *Leuconostoc gasicomitatum*, *Lactobacillus brevis*, *Lactobacillus curvatus*, *Leuconostoc citreum*, *Pediococcus pentosaceus*, *Lactobacillus sakei*, *Weissella confusa*, and *Leuconostoc mesenteroides*
Cucumber	Khalpi	*Leuconostoc fallax*, *Lactobacillus brevis* and *Lactobacillus plantarum*
Cucumber	Jiang-guais	*Enterococcus casseliflavus*, *Weissella hellenica*, *Leuconostoc lactis*, *Lactobacillus Plantarum* and *Weissella cibaria*
Cummingcordia	Pobuzihi	*Weissella cibaria*, *Pediococcus pentosaceus*, *Lactobacills plantarum*, *Lactobacillus pobuzihii* and *Weissella paramesenteroides*
Durian fruit	Tempoyak	*Lactobacills durianis Lactobacillus brevis Leuconostoc mesenteroides Lactobacillus fermentum* and *Liquorilactobacillus mali*
Field mustard	Nozawana-zuke	*Leuconostoc and Lactobacillus*
Fresh cabbage	Sauerkraut	*Lactobacillus* spp., *Leuconostoc* spp. and *Pediococcus* spp.
Fresh peaches	Yan-taozih	*Weissella cibaria*, *Lactobacillus brevis*, *Weissella minor*, *Leuconostoc mesenteroides, Enterococcus faecalis*, *Lactobacillus lactis* and *Weissella paramesenteroides*
Ginger	Yan-jiangis	*Lactobacillus plantarum* and *Weissella cibaria*
Grapes	Wine (red)	*Lactobacillus Plantarum*, *Pediococcus parvulus*, *Oenococcus oeni* and *Lactobacillus casei*
Green peppers and green tomatoes	Tursu	*Pediococcus pentosaceus*, *Leuconostoc mesenteroides*, *Lactobacillus brevis* and *Lactobacillus plantarum*
Maganesaag	Goyang	*Lactobacillus Brevis*, *Pediococcus pentosaceus*, *Lactococcus lactis*, *yeasts Candida* spp., *Enterococcus faecium* and *Lactobacillus plantarum*
Mustard leaves	Inziangsang	*Pediococcus Lactobacillus plantarum* and *Lactobacillus brevis*
Mustard cabbage leaf	Sayur asin	*Lactobacillus confusus*, *Lactobacillus plantarum*, *Leuconostoc mesenteroides* and *Pediococcus pentosaceus*
Rayosag, mustard leaves, cauliflower leaves, and cabbages	Gundruk	*Pediococcus pentosaceus*, *Lactobacillus casei*, *Lactobacillus plantarum* and *Lactobacillus fermentum*
Radish taproot	Sinki	*Lactobacillus casei*, *Leuconostoc fallax*, *Lactobacillus brevis* and *Lactobacillus plantarum*
Turnips	Shalgam juice	*Lactobacillus paracasei*, *Pediococcus pentosaceus*, *Lactobacillus brevis* and *Lactobacillus buchneri*
Wax gourd	Yan-Dong-Gua	*Weissella cibaria* and *Weissella paramesenteroides*

**Table 2 pharmaceutics-14-02543-t002:** Probiotic compositions, indicating their pharmacological activity and their outcomes.

Probiotic Strain	Assay	Results	References
**Antioxidant effect**			
*Bacillus amyloliquefaciens*, *Starmerella bombicola*, and *Lactobacillus brevis*	DPPH, ABTS	ABTS antioxidant activity tests of *Bacillus amyloliquefaciens* (400 µg/mL) showed 1.01-, 1.03- and 1.05-fold increases in antioxidant activity in comparison to *Lactobacillus brevis*, *Starmerella bombicola* and blueberry fruit extract without probiotic bacteria A DPPH radical assay revealed that *Bacillus amyloliquefaciens* (1600 µg/mL) led to an increase in antioxidant activity by 1.01-, 1- and 1.23-fold as compared to Lactobacillus brevis, Starmerella bombicola, and blueberry fruit extract without probiotic bacteria	[14]
*Bifidobacterium breve*, *Rhamnosus GG*, *Probionebacterium freudenreichii* and *Lactobacillus retueria*,	DPPH, ABTS	A DPPH antioxidant scavenging assay revealed that *Probionebacterium freudenreichii (*100 µg/mL) strain led to 1.01-, 1.12-, 1.06-, 1.05- and 1.04-fold increases in antioxidant activity in comparison to *Lactobacillus retueria*, *Bifidobacterium breve* and *Lactobacillus rhamnosus*, ascorbic acid, and butylated hydroxytoluene ABTS antioxidant activity tests of *Probionebacterium freudenreichii (* (100 µg/mL) strain revealed an increase in antioxidant activity by 1-, 1-, 1.06-, 1.01- and 1.01-fold as compared to *Lactobacillus rhamnosus*, *Lactobacillus retueria*, *Bifidobacterium breve*, ascorbic acid, and Butylated hydroxytoluene	[15]
BS1, BS2, BV	TAOC, MDA, SOD	TAOC results revealed that BV led to 1.17-, 1.11- and 2.5-fold increase in antioxidant activity in comparison to BS2, BS1and saline-treated group (Control) MDA study: BS2 treated groups showed 3.6-, 1.05- and 1.11-fold decreases in MDA level as compared to control, BS1 and BV1 treated groups SOD study showed that BS2 treated groups exhibited an increase in antioxidant activity by 1.7-, 1.2- and 1.4-fold in comparison to control, BS1 and BV1 treated groups	[16]
*Enterococcus faecium*	DPPH, Superoxide, Hydroxyl scavenging assay	DPPH assay showed that *Enterococcus faecium* (10 mg/mL) led to a 1.08-fold increase in antioxidant activity as compared to ascorbic acid Superoxide scavenging assay revealed *Enterococcus faecium* (10 mg/mL) led to a 1.13-fold increase in antioxidant activity in comparison to ascorbic acid Hydroxyl scavenging assay result revealed that *Enterococcus faecium* (10 mg/mL) led to a 1.42-fold in antioxidant activity as compared to ascorbic acid	[17]
*Lactobacillus acidophilus*	DPPH	SY (0.2 mg/mL) led to a 1.16-, 1- and 1.04-fold increase in antioxidant activity in comparison to control, SWY and WY, respectively	[9]
*Lactobacillus plantarum*, *Lactobacillus rhamnosus*, *Lactobacillus casei*,	DPPH	DPPH assay revealed that *Lactobacillus* rhamnosus (0.1 mg/mL) led to a 1.21-, 1.19- and 1.46-fold increase in antioxidant activity as compared to *Lactobacillus casei, Lactobacillus plantarum* and cashew milk-yoghurt without probiotic strain	[18]
*Lactobacillus plantarum* DM5	DPPH, Superoxide anion, Hydroxyl	*Lactobacillus plantarum* DM5 (10^10^ CFU/mL) has 20% and 30% higher hydroxyl radical activity than *Lactobacillus acidophilus and Lactobacillus plantarum* *Lactobacillus plantarum* DM5 (10^10^ CFU/mL) showed 31% and 22% higher superoxide anion scavenging activity than *Lactobacillus Plantarum* and *Lactobacillus acidophilus* *Lactobacillus plantarum* DM5 (10^10^ CFU/mL) exhibited an increase in DPPH scavenging activity by 43% and 33%, as compared to *Lactobacillus plantarum* and *Lactobacillus acidophilus*	[19]
*Lactobacillus paracasei* A-4, *Lactobacillus plantarum* A-7, *Lactobacillus paracasei* BL-12, *Lactobacillus paracasei* DU-8, *Lactococcus lactis* T-8	DPPH	*Lactobacillus plantarum* A-7 1 mg/mL) exhibited increase in antioxidant activity by 1.22-, 2.81-, 3.19-, 1.01-, 3.47- and 5.41-fold as compared to *Lactobacillus paracasei* A-4, *Lactobacillus paracasei* BL-12, *Lactobacillus paracasei* DU-8, *Lactobacillus brevis* O-9, *Lactococcus lactis* T-8 and Control milk respectively	[20]
**Anti-inflammatory**			
**Probiotic strain**	**Design/** **participants**	**Results**	**References**
*Bifidobacterium animalis* ssp. *lactis 420* (900 billion CFU/day)	Randomized/50	Improved bacterial dysbiosis and immunity Reconstructed the balance of intestinal flora	[21]
*Lactobacillus acidophilus La-5* and *Bifidobacterium BB-12* (10^6^ CFU/g each)	Randomized double-blind/210	Decreased inflammation Increased bacterial count in the intestine and colon	[22]
*Lactobacillus acidophilus*, *Lactobacillus casei*, *Bifidobacterium bifidum*, *Lactobacillus fermentum* (2 × 10^9^ CFU/g each)	Randomized double-blind/48	Improved glucose homeostasis. Decreased oxidative stress and inflammation	[23]
*Lactobacillus acidophilus*, *Lactobacillus infantis*, *Bifidobacterium bifidum*, *Lactobacillus fermentum* and *Bifidobacterium longum* (6 billion CFU each)	Randomized double-blind/ 52	Decreased proinflammatory mediators of inflammation	[24]
*Lactobacillus plantarum OLL2712* (5 × 10^9^ CFU)	Randomized/ 130	Decreased chronic inflammation Decreased HbA1c level	[25]
**Immunomodulatory effect**			
**Probiotics strain**	**Animal model/other**	**Results**	**References**
*Bifidobacterium longum KACC 91563*(100 billion CFU/g)	Male BALB/c mice	Improved systemic immunity Regulated T and B-cell proliferation Inhibited the Th1cytokine imbalance and immune cytokine production	[26]
*Bifidobacterium longum* CCUG 52486 (5 × 10^8^ CFU/day)	Human	Increased NK cell activity Increased the number of IgG^+^ memory B-cells	[27]
*Lactobacillus casei* Shirota (1.3 × 10^10^ CFU/day)	Human	Increased innate immunity by increasing levels of natural killer cell activity Increased inflammatory status by promoting IL-10/IL-12 ratio	[28]
*Lactobacillus casei*; CRL 431 (10^9^ cells/day)	Female BALB/c mice	Increased mucosal activity Maintain homeostasis at the mucosal level Increased phagocytosis Increased IL-10 levels	[29]
*Limosilactobacillus fermentum* (10^9^ CFU/mL)	Female Balb/c mice	Modulated inflammatory cytokines Stimulated response of the immune system	[30]
**Antidiabetic effect**			
**Probiotic strain**	**Animal model**	**Results**	**References**
*Lactobacillus* casei (4.0 × 10^9^ CFU/rat/day)	Rat	↓BGL	[31]
*Lactobacillus casei* and *Bifidio bifidum* (1 × 10^7^ cfu/mL)	Wistar rat	↓ BGL, ↓ HbA1c, ↓ TC, ↓ TGs ↓ LDL, ↓ VLDL, ↑ HDL	[32]
*Lactobacillus.casei* (10^9^ CFU/mL)	Mice	↓ BGL, ↓ insulin ↓ insulin-like growth factor I, ↓ C-peptide	[33]
*Lactobacillus casei CCFM419* (10^9^ CFU)	Mice	↓ Fasting and postprandial blood glucose ↓ glucose intolerance, ↓ IR, ↓ TNFα, ↓ IL-6, ↑ GLP-1	[34]
*Lactobacillus. Gasseri* (6 × 10^7^ cfu/g)	Rat	↓ BGL, ↓ IR, ↓ inflammation ↑ SCFA, ↑ insulin secretion	[35]
*Lactobacillus plantarum CCFM0236* (8 × 10^9^ cfu/mL)	Mice	↓ Food intake, ↓ BGL, ↓ HbA1c, ↓ leptin level, ↓ insulin level ↓ TNFα, ↓ HOMA-IR index, ↑ activities of GPx	[36]
*Lactobacillus.plantarum, strain Ln4* (5 × 10^8^ cfu/day)	Male mice	↓ Weight gain, ↓ epididymal fat mass, ↓ total plasma TG level ↓ HOMA-IR, ↑ glucose tolerance, ↑ insulin response	[37]
*Lactobacillus.plantarum MTCC5690* and *Lactobacillus fermentum MTCC5689* (1.5 × 10^9^ colonies/day)	C57BL/6J male mice	↓ IR, ↓ glucose intolerance, ↓ glucose level, ↓ lipid level, ↓ TNFα ↓IL6 ↑ gene expression patterns of intestinal tight junction	[38]
*Lactobacillus.rhamnoss*, *Lactobacillus.acidophilus*, *Bifidio bifidumi* (6 × 10^8^ CFU each)	Mice	↓ Intestinal permeability, ↓ LPS translocation, ↓ low-grade systemic inflammation ↓ glucose tolerance, ↓ hyperphagic behavior, ↓ hypothalamic insulin, and leptin resistance	[39]

ABTS 2,2′-azino-bis(3-ethylbenzothiazoline-6-sulfonic) acid, CFU/g; Colony forming units/gram, TAOC; Total antioxidant capacity, MDA; maleic dialdehyde, GSH-PX; Glutathione peroxidase, SOD; Superoxide dismutase, BS1; Bacillus subtilis1, BS2; Bacillus subtilis2, BV; Bacillus velezensisis, SY; Probiotic fat-free yogurt, SWY; Probiotic semi-fat yogurt, WY; Probiotic full fat yogurt; DPPH; 2,2-DiPhenyl-2-Picryl hydrazyl hydrate. Here sign ↓ indicates decrease in the level and ↑ indicates increase in the level.

**Table 3 pharmaceutics-14-02543-t003:** Different stabilization techniques for probiotics.

Probiotic Strains	Microencapsulation Technique	Parameters Test	Observation	References
LA and BL	Spray chilling	Viability count	Stability of probiotics was enhanced for 4 months A microencapsulated blend of probiotics containing BL and LA exhibited a 5.2-fold increase in cell viability on the 120th day as compared to non-encapsulated probiotics blend	[56]
LRIMC-501	Spray chilling	Viability count	The blend of probiotics showed stability of LRIMC-501 for 12 months Microencapsulated LR IMC 501 exhibited 100-fold increase in cell viability as compared to its non-encapsulated form	[72]
Ls	Spray coating using Sucrose	Viability count	Stability of probiotics was enhanced for 24 months The sucrose coating improved the bacterial viability by 4.28-fold as compared to non-coated probiotics blend	[73]
LA	Spray coating using maize and potato	Viability count	Stability of probiotics was enhanced for 42 days Maize coated probiotics exhibited an increase in cell viability by 1.11-fold and 1.03-fold as compared to non-encapsulated and rice coated probiotics	[74]
LA	Fluidized bed coating	Thermal stability	Fluidized bed coated probiotics showed a 6.3-fold increase in cell viability at 90 °C for 30 min as compared to non-coated probiotics	[75]
LS	Fluidized bed coating	Thermal stability	Fluidized bed-coated probiotics showed a 15.22% increase in cell viability as compared non-encapsulated probiotics	[76]
LA	Liposome	Thermal stability	A probiotic blend was able to bear a thermal stress of 50 °C Surface layer protein-based liposomes exhibited 1.56-fold decrease in carboxyfluorescein leakage as compared to control liposomes	[77]
LP-PR01	Extrusion-dripping technique	Thermal stability	Encapsulated probiotics showed greater stability than non-encapsulated probiotics at 4 °C	[78]
LA-*ATCC-4356*	Extrusion-dripping technique	Thermal stability	Encapsulated probiotics exhibited higher cell viability at 65 °C as compared to non-encapsulated probiotics The encapsulation of probiotics prolonged their shelf life up to 15 days	[79]
*Enterococcus*	Spray drying	Stability	Spray drying protected probiotics against degradation from bile salts Stability of probiotics was enhanced for 60 days Spray dried probiotic powder exhibited a 2.56-fold increase in cell viability at 4 °C as compared to non-coated probiotic powder kept at room temperature	[80]
ST IFFI 6038	Extrusion	Viability count	Extrusion-based probiotic microcapsules exhibited a 3.5-fold increase in viable count as compared to ST IFFI 6038 powder	[81]
LP	pH induced gelation	Viability count	LP microencapsulated probiotics exhibited 1.14-fold increase in cell viability within 21 days as compared to non-encapsulated probiotics	[82]
Ls	Alginate coating by homogenization pressure	Viability count	Microencapsulated probiotics exhibited 1.1-fold increase in cell viability as compared to non-encapsulated probiotics	[83]
LB-ST-69	Matrix polymerization	Viability count	Microencapsulated probiotics exhibited 1.26-fold increase in cell viability as compared to non-encapsulated probiotics At room temperature microencapsulated probiotics showed 1.31-fold increase in cell survival rate as compared to non-encapsulated probiotics within 28 days	[84]
YEP	Co-extrusion	Viability count	Encapsulated probiotics exhibited 1.8-fold increase in cell1viability as compared to non-encapsulated probiotics at 4 °C	[85]

BL; Bifibobacterium lactis, LA; Lactobacillus acidophilus, LB-ST-69; Lactobacillus brevis ST-69, LP; Lactobacillus paracasei, Ls; Lactobacillus salivarius, LS; Lactobacillus sporogenes, ST-IFFI-6038; Streptococcus thermophilus IFFI 6038, LA-ATCC-4356; Lactobacillus acidophilus ATCC-4356, LP-PR01; Lactobacillus pentosus PR01, YEP; Yeast extracted probiotics.

**Table 4 pharmaceutics-14-02543-t004:** List of commercialized probiotics as nutraceutical.

Brand and Trade Name	Manufacturer	Country	Stains Isolated	Food Type	References
Aciforce	Biohorma	The Netherlands	*Enterococcus faecium*, *Lactobacillus acidophilus*, *Bifidobacterium bifidum*, *Lactococcus lactis*	Lyophilized products	
Activia	Danone	France	*Bifidus actiregularis*	Creamy yoghurt	
Actimel	Danone	France	*Lactobacillus casei Immunitas*	Probiotic yoghurt drink	
Bacilac	THT	Belgium	*Lactobacillus acidophilus*, *Lactobacillus rhamnosus*	Lyophilized product	
Bactisubtil	Synthelabo	Belgium	*Bacillus* sp. *strain IP5832*	Lyophilized product	
Hellus	Tallinna Piimatööstuse AS	Estonia	*Lactobacillus fermentum ME-3*	Dairy product	
Jovita Probiotisch	H & J Bruggen	Germany	*Lactobacillus strain*	Probiotic yoghurt	[88]
Proflora	Chefaro	Belgium	*Lactobacillus delbrueckii* subsp. *bulgaricus*, *Lactobacillus acidophilus*, *Bifidobacterium*, *Streptococcus thermophilus*	Lyophilized product	
Provie	Skanemejerier	Sweden	*Lactobacillus plantarum*	Fruit drink	
ProViva	Skanemejerier	Sweden	*Lactobacillus plantarum*	Fruit drink	
Rela	Ingman Foods	Finland	*Lactobacillus reuteri*	Cultured milk	
Revital Active	Olma	Czech Republic	*Lactobacillus acidophilus*	yoghurt drink	
Yakult	Yakult	Japan	*Lactobacillus casei* Shirota	Milk drink	
Yosa	Bioferme	Finland	*Bifidobacterium lactis*, *Lactobacillus acidophilus*	Yoghurt-like oat product	
Vitamel	Campina	The Netherlands	*Lactobacillus casei* GG, *Lactobacillus acidophilus*, *Bifidobacterium bifidum*	Dairy products	
Vifit	Campina	The Netherlands	*Lactobacillus strain*	Yoghurt drink	
Activia	Danone	France	*Bifidus actiregularis*	Creamy yoghurt	

**Table 5 pharmaceutics-14-02543-t005:** List of probiotics under clinical investigation.

Probiotic Name	Manufacturer	Strain	Colony Forming Units (CFUs)	Health Claims	References
Activa yogurt	Dannon Inc	*Lactobacillus bulgaricus*, *Streptococcus thermophilus*, *Bifidobacterium regularis*, *Bifidobacterium animalis* DN-173010	10 billion	Antibacterial activity Lipid lowering activity Maintain gut microflora	[89]
Adult Formula CP-1	Custom Probiotics Inc	*Lactobacillus rhamnosus*, *Lactobacillus acidophilus*, *Bifidobacterium bifidum*, *Bifidobacterium lactis*	50 billion	Immunomodulatory effect Maintain gut microflora Antibacterial activity Improve pancreatitis Lipid lowering action	[90]
Align capsules	Proctor & Gamble	*Bifidobacterium. infantis* 35624	1 billion	Increased immunity	[91]
Attune nutrition bars	Attune Foods	*Lactobacillus case*i Lc-11, *Bifidobacterium lactis* HN019, *Lactobacillus acidophilus* NCFM	6.1 billion	Antitumor activity	[92]
Bio-K+ cultured milk-based probiotic	Bio-K+ Int Inc.	*Lactobacillus case*i LBC804, *Lactobacillus acidophilus* CL1285	50 billion	Antibacterial activity	[93]
Bio-K+ probiotic capsules	Bio-K+ Int Inc.	*Lactobacillus casei* LBC804, *Lactobacillus acidophilus* CL1285	50 billion	Antibacterial activity	[94]
Culturelle capsules	Amerifit Nutrition, Inc	*Lactobacillus rhamnosus* GG	10 billion	Immunomodulatory effect Activity against toxins Inhibit reactive oxygen species Action against inflammatory bowel disease	[95]
Gefilus juice	Valio Ltd.	*Lactobacillus rhamnosus* GG	5 million	Immunomodulatory effect Activity against toxins Inhibit reactive oxygen species	[96]
Gerber Good Start Protect Plus powdered infant milk formula	Nestle	*Bifidobacterium lactis* Bb-12	10 billion	Anticancer effect Maintain gut microflora	[97]
Good Belly fruit drink	Next Foods	*Lactobacillus plantarum* 299v	20 billion	Antimicrobial action Improve pancreatitis	[98]
OWP probiotics	One Wellness Place	*Bifidobacterium breve, Bifidobacterium longum*, *Bifidobacterium infantis*, *Lactobacillus acidophilus*, *Lactobacillus plantarum*, *Lactobacillus rhamnosus*	15 billion	Immunomodulatory effect Maintain gut microflora Action against inflammatory bowel disease Antibacterial activity Improve pancreatitis	[99]
Ultimate Probiotic Formula	Swanson Health Products	*Bifidobacterium longum*, *Bifidobacterium lactis*, *Lactobacillus plantarum*, *Lactobacillus case*i, *Lactobacillus sylvarius*, *Lactobacillus bulgaricus*, *Lactobacillus sporogenes* + Prebiotic NutraFlora FOS	60 billion	Immunomodulatory effect Maintain gut microflora Action against inflammatory bowel disease Antibacterial activity Improve pancreatitis Improve arthritis	[100]
VSL#3 saket	Sigma-Tau Pharmaceuticals	*Bifidobacterium breve, Bifidobacterium longum*, *Bifidobacterium infantis*, *Lactobacillus acidophilus*, *Streptococcus thermophilus*, *Lactobacillus casei*	450 billion	Lipid lowering action Improve pancreatitis Antibacterial activity Action against inflammatory bowel disease	[101]
Yo-Plus yogurt	Yoplait Inc	*Bifidobacterium animalis* subsp Bb-12, *Streptococcus thermophilus*, *Lactobacillus bulgaricus* _+_ Prebiotics	>5 billion	Immunomodulatory effect Maintain gut microflora Action against inflammatory bowel disease Antibacterial activity Improve pancreatitis	[102]

**Table 6 pharmaceutics-14-02543-t006:** List of various patents filed on probiotics.

Probiotic Formulation Composition	Patent Number	Beneficial Claims	References
**Therapeutic potential**			
A61K35/741—Probiotics	WO2019180748A1	Immunomodulatory, antibacterial and anti-inflammatory action	[103]
*Bacillus circulans* ATCC PTA-5614, 5615, 5616	US 7361497 B2	Treat Salmonellosis in food production animals	[104]
*Bacillus* strain, *Saccharomyces cerevisiae*, *Saccharomyces boulardii*, LAB	US20180280312A1	Enhance stability and antibacterial action at wound site	[105]
*Bacillus subtilis*, *Lactobacillus plantarum*	RU2401116C2	Treatment of burn related wounds and antibacterial action	
*Bifidobacterium* strain AH1714	CN102946891A	Immunomodulatory effect	[106]
*Enterococcus faecium*	EP0508701A2	Treat inflammatory bowel disease	[107]
*Enterococcus mundtii*	KR20090023626A	Antibacterial activity	[108]
*Lactobacillus acidophilus LPV 31*	EP2450062A1	Treat burn and ulcer related wounds	[109]
LAB	KR101885403B1	Antimicrobial activity against *Pseudomonas aeurogonisa* and *Staphylococcus aureus*	[110]
*Lactobacillus casei*, *Lactobacillus* *rhamnosus* + tagatose	EP2837292 A1	Increase growth of *Lactobacillus spp*. in the intestine	[111]
*Lactobacillus* genera, *Bifidobacterium* genera	US20030017192 A1	Improve gut dysbiosis	[112]
*Lactobacillus plantarum*, *Lactobacillus brevis*	KR102083002B1	Ensure probiotic stability and provide wound healing	[113]
*Lactobacillus plantarum*, *Lactobacillus acidophilus*	WO2020261055A1	Re-epithelization and antibacterial action	[114]
*Lactobacillus plantarum*, *Lactobacillus acidophilus*, *Bifidobacterium longum*	JP6944399B2	Wound healing action	[115]
Probiotic bacteria + sodium laureth sulfate + alkyl polyglycozide + cocamide DEA + glycerol + orange terpenes + fragrance + D-pantenol + ethyl hydroxy ethyl cellulose + orange terpenes + citric acid	WO2017099559A1	Increase stability and the survival rate of probiotic strain	[116]
Probiotic + valproic acid	US20190282523A1	Treat acne, wounds and MRSA infections	[117]
Recombinant probiotic	CN107438666B	Treatment of inflammatory skin dysfunction	[118]
**Nutraceutical**			
*Bacillus coagulans*, clostridium, *Bacillus subtilis* or *Lactobacillus sporogenes* + arabinogalactan	EP1607096B1	Increase the colonization of gut microflora	[119]
Bifidobacterium, Lactococcus and Staphylococcus, Saccharomyces, Clostridium, Lactobacillus, Enteroccus, Peptostreptococcus, Eubacterium, Streptococcus,	WO 1996008261 A1	Provides health benefits	[120]
*Bifidobacterium longum*, *Bfidobacterium bifidum*, *Lactobacillus salivarius*, *Lactobacillus acidophilus*, *Bifidobacterium infantis*, L-glutamine, fructooligosaccharides and N-acetyl glucosamine	US6468525B1	Maintain the gut microflora	[121]
Probiotic food	WO2002065840A3	Improve stability and make them as a consumable product	[122]

## Data Availability

Not applicable.

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
