# Peer review of "Gut Dysbiosis and Diabetic Foot Ulcer: Role of Probiotics"

_pharmaceutics, 2022, doi:10.3390/pharmaceutics14112543_

Round 1

Reviewer 1 Report

The review summarizes the role of probiotics in the treatment of DM/DFU. After reviewing the whole review article, there are some comments as follows:

1.       Authors should pay more attention to details. All abbreviations should be given their full names when they first appear. For example, the full name of "DW" and “DM” first appeared in the abstract.

2.       The “A”, “B”, and “C” is missing in Figure 1.

3.       The caption of Table 2 should be written.

4.       What is the concentration of probiotics used in each biological assay. It should be summarized in tables.

5.       What are the advantages and disadvantages of each technique for the stabilization of probiotics. 

Author Response

1. Authors should pay more attention to details. All abbreviations should be given their full names when they first appear. For example, the full name of "DW" and “DM” first appeared in the abstract.

Response: Corrected as suggested

2. The “A”, “B”, and “C” is missing in Figure 1.

Response: Corrected as suggested

3. The caption of Table 2 should be written.

Response: Added as per the suggestion of learned reviewer.

4. What is the concentration of probiotics used in each biological assay. It should be summarized in tables.

Response: Concentration is now mentioned in the Table

5.   What are the advantages and disadvantages of each technique for the stabilization of probiotics. 

Response: I thank the learned reviewer for the valuable suggestions. We have added advantages and disadvantages of probiotics in section 6.

Reviewer 2 Report

Overall a good job but needs some clarification

Row:

- 70: also offloading was a therapeutic treatment for neuropatic ulcer

- 72: treatment strategies to correct DFU, could be associated to postoperative complication but these are not all due to poor control of hyperglycemia. Specifies this statement.

-212: Were other therapies such as antibiotics or compression bandages done?

-208/222: Explains the choice and rationale for the use of topical and non-systemic probiotics

-232: Grade 3…what is the scale? Wagner, Texas university…? And what type of ulcers were considered? Vascular, neuropathic or neuro-ischemic? And what type of treatment was done?

-285: “every eighth day”, there was administered the drug at the day one and eight? Basically piperacillin/tazobactam was administered tid.

-294: explain that this patient was also subjected to percutaneous transluminal angioplasty (PTA) with drug eluting balloon (DEB) ranger 5 × 100 mm in superficial femoral artery (SFA) and right popliteal artery, followed by surgical curettage of necrotic forefoot injuries and amputation of the second toe of the right foot.

-299: explain that systemic or topical antibiotic treatment was not administered during the course of probiotics.

-409: explain that use of  probiotics had remarkable improvement on DFU but clarify what is the better way of administration (for os or locally)

-412: limited to in vitro ed animal study mainly

Author Response

1          Also offloading was a therapeutic treatment for neuropathic ulcer  

Response: Comment addressed in Line 71.

2          Treatment strategies to correct DFU could be associated to postoperative complication but these are not all due to poor control of hyperglycemia. Specifies this statement.

Response: Comment addressed in the Introduction section Line 77-80

3          Were other therapies such as antibiotics or compression bandages done?   

Response: The other approaches used in DFU are added in the line 71-77

4          Explains the choice and rationale for the use of topical and non-systemic probiotics Response: Comment incorporated in the section 8 conclusion line 450-454

5          What is the scale? Wagner, Texas university…? And what type of ulcers were considered?  Vascular, neuropathic or neuro-ischemic? And what type of treatment was done?

Response: Comment addressed in the Venosi et al study line 247

All these are mentioned in the study and highlighted with yellow with track changes

6          Every eighth day”, there was administered the drug at the day one and eight? Basically piperacillin/tazobactam was administered tid.          

Response: Comment addressed in the line 299-300

7          Explain that this patient was also subjected to Percutaneous transluminal angioplasty (PTA) with drug eluting balloon (DEB) ranger 5 × 100 mm in superficial femoral artery (SFA) and right popliteal artery, followed by surgical curettage of necrotic forefoot injuries and amputation of the second toe of the right foot.           

Response: Comment is addressed in the Venosi et al. (2019) study line 294-298

  1. Explain that systemic or topical antibiotic treatment was not administered during the course of probiotics.

Response: The comment is addressed in the line 310-311

  1. Explain that use of probiotics had remarkable improvement on DFU but clarify what is the better way of administration (for os or locally)

Response: The comment is incorporated in section 8 of conclusion line 449-455

  1. Limited to in vitro and animal study mainly

Response: Comment is addressed in the conclusion section line 445-460